# Genome-Wide Identification and Expression Analysis of *AS2* Genes in *Brassica rapa* Reveal Their Potential Roles in Abiotic Stress

**DOI:** 10.3390/ijms241310534

**Published:** 2023-06-23

**Authors:** Qiwei Jiang, Xiaoyu Wu, Xiaoyu Zhang, Zhaojing Ji, Yunyun Cao, Qiaohong Duan, Jiabao Huang

**Affiliations:** College of Horticulture Science and Engineering, Shandong Agricultural University, Tai’an 271000, China

**Keywords:** *Brassica rapa*, *AS2*, bioinformatics analysis, expression profile, abiotic stress

## Abstract

The *ASYMMETRIC LEAVES2/LATERAL ORGAN BOUNDARIES* (*AS2/LOB*) gene family plays a pivotal role in plant growth, induction of phytohormones, and the abiotic stress response. However, the *AS2* gene family in *Brassica rapa* has yet to be investigated. In this study, we identified 62 *AS2* genes in the *B. rapa* genome, which were classified into six subfamilies and distributed across 10 chromosomes. Sequence analysis of *BrAS2* promotors showed that there are several typical *cis*-elements involved in abiotic stress tolerance and stress-related hormone response. Tissue-specific expression analysis showed that *BrAS2-47* exhibited ubiquitous expression in all tissues, indicating it may be involved in many biological processes. Gene expression analysis showed that the expressions of *BrAS2-47* and *BrAS2-10* were significantly downregulated under cold stress, heat stress, drought stress, and salt stress, while *BrAS2-58* expression was significantly upregulated under heat stress. RT-qPCR also confirmed that the expression of *BrAS2-47* and *BrAS2-10* was significantly downregulated under cold stress, drought stress, and salt stress, and in addition *BrAS2-56* and *BrAS2-4* also changed significantly under the three stresses. In addition, protein–protein interaction (PPI) network analysis revealed that the *Arabidopsis thaliana* genes *AT5G67420* (homologous gene of *BrAS2-47* and *BrAS2-10*) and *AT3G49940* (homologous gene of *BrAS2-58*) can interact with NIN-like protein 7 (NLP7), which has been previously reported to play a role in resistance to adverse environments. In summary, our findings suggest that among the *BrAS2* gene family, *BrAS2-47* and *BrAS2-10* have the most potential for the regulation of abiotic stress tolerance. These results will facilitate future functional investigations of *BrAS2* genes in *B. rapa*.

## 1. Introduction

The *ASYMMETRIC LEAVES2/LATERAL ORGAN BOUNDARIES* (*AS2/LOB*) gene family, equivalent to the *LBD* gene family, is a plant-specific transcription factor family that was first discovered in *Arabidopsis thaliana* in 2002 through enhancer trap insertion and that has since received considerable attention [1,2]. Members of the *AS2/LOB* gene family govern the development and morphology of plant leaves and play a pivotal role in the development of lateral organs, such as the leaves, petals, and fruits. Members of the *AS2* gene family interact with other transcription factors to regulate gene expression in plants, thereby affecting plant development and morphology [1]. The *AS2* gene family is highly conserved, encoding three conserved domains: the CX2CX6CX3C motif (LOB domain), which is composed of four conserved cysteine residues, conserved glycine residues, and the LX6LX3LX6L leucine-like zipper motif. The *AS2* gene family has been identified in many plants. Among them, there are 43 members in *A. thaliana*, 24 members in *Hordeum vulgare*, 57 members in *Medicago truncatula*, 50 members in *Vitis vinifera,* 43 members in *Solanum tuberosum*, and 18 members in *Glycine max* [1,3,4,5,6,7].

The prominent contribution of the *AS2* gene family to the plant defense response against abiotic stress has been studied. A low expression of *StLBD1-5* promoted resistance to drought conditions in *S. tuberosum* [6]. Overexpressing *PheLBD29* in *Moso bamboo* enhanced drought stress tolerance in transgenic plants [8]. Additionally, the *LBD15* loss-of-function mutant in *A. thaliana* decreased sensitivity to abscisic acid (ABA) and increased sensitivity to water-deficit stress [9]. Overexpressing *MtLBD1* (an ABA and salt-response transcription factor) in *M. truncatula* showed that ABA mediates the response to salt stress [10]. *ZmLBD5* was shown to be a negative regulator of drought tolerance in *Zea mays*, based on the phenotype of CRISPR/Cas9 knockout *LBD5* seedlings under drought stress. Modulating ABA biosynthesis is a potentially useful target for growth response to water deficit [11]. Investigating *AtHB12* mutant (which mediates the growth response to water deficit) and *LBD14 RNAi* plants demonstrated that ABA mediates the inhibition of the formation of lateral roots via a non-auxin-dependent pathway in response to abiotic stress [12]. In *Solanum lycopersicum*, studies on the *SlLBD40* knockout system indicated that it is a negative regulator of drought tolerance through the jasmonic acid (JA) signal transduction pathway [13]. Furthermore, *AS2* genes are involved in modulating pathogen and pest resistance [14]. In *A. thaliana* and *Physcomitrella patens, LBD20* and *PpLBD27* are related to disease susceptibility and pathogens [15,16]. Moreover, low temperature, NaCl, drought, and other abiotic stresses promote the production of secondary metabolites, so plants can adapt to their environment, and there may be a “bridge” between the transcription factor, the biosynthetic gene, and the secondary metabolite mechanisms by which *LBD* genes control secondary metabolism [17,18,19,20]. In *A. thaliana*, *AtLBD37*, *AtLBD38*, and *AtLBD39* genes act as repressors of anthocyanin synthesis and as N availability signals [21,22]. In *Oryza sativa*, *OsLBD37* has been linked to nitrogen metabolism and exhibits responses to various environmental stresses [21,23]. In *Camellia sinensis*, overexpression of *CsLBD39* revealed that it may be involved as a negative regulator of nitrate signal transduction in tea plants [24].

*AS2* also plays essential roles in growth and development, as it is expressed in the adaxial section of leaf primordia and young flower organs, where it participates with *AS1* and *JAGGED* (*JAG*) to impose limitations on the boundary cells in flower organs [25,26]. *AS2* encodes a nucleoprotein with a plant-specific AS2/LOB domain that inhibits *KNOX* expression, which is necessary for the differentiation of the primordia [27,28,29,30]. Expression of the homeobox *KNOX* gene in *A. thaliana* must be retained in the shoot meristem for it to remain active. Genetic analysis has shown that *AS1* and *AS2* genes play a crucial role in sepal and petal primordia, where they inhibit boundary-specific genes, to promote the proper development of organs [31,32]. Similarly, studies in *Brassica campestris* have shown that *BcAS2* forms a complex with *BcAS1-1/2*, to establish the paraxial abaxial polarity of the lateral organs [33]. Overall, the function of *AS2* is crucial in ensuring proper plant development and organ differentiation.

*Brassica rapa* (Chinese cabbage) is a widely grown vegetable in Asia that is often exposed to various stresses during growth. However, no study has examined the *AS2* genes in *B. rapa*, until now. In this study, the *BrAS2* genes were identified at the whole-genome level, and the physicochemical properties, structures of the genes and proteins, and expression profiles were analyzed. This study provides basic information that will aid future functional studies on *BrAS2s*.

## 2. Results

### 2.1. Genome-Wide Identification and Physicochemical Characterization of the Brassica rapa AS2 Genes

To identify and understand the characteristics of the *AS2* gene family in *B. rapa*, we used bioinformatics methods to analyze the *B. rapa* genome, ultimately resulting in the discovery of 62 members within the *BrAS2* gene family through homology matching with the *B. rapa* genomic database (Table 1). These genes were renamed *BrAS2-1* to *BrAS2-62* and were further categorized into six distinct subfamilies. The physicochemical properties of the *BrAS2* genes were analyzed and are listed in Table 1. The amino acid (aa) length of the BrAS2 proteins ranged from 107 to 394 aa; the molecular weight was 44,228.21, while the minimum was 11,876.75, and the PI values ranged from 4.5 to 10.63.

### 2.2. Phylogenetic Relationships and Synteny Analysis

According to the gff3 genome annotations, 62 *BrAS2s* were mapped to the *B. rapa* chromosomes. The *BrAS2* genes were extensively dispersed across the 10 chromosomes, as depicted in Figure 1A. Chromosomes 4 and 9 contained the highest density of genes, encompassing 10 genes that constituted 16.1% of all *BrAS2* genes, while chromosome 8 harbored only one gene, accounting for 1.61% of all *BrAS2* genes. The subclasses were widely and sporadically distributed across the 10 chromosomes, without any particular concentration on a specific chromosome. Additionally, to elucidate the phylogenetic relationships between the *BrAS2* genes, we conducted an intraspecific collinearity analysis and found that 5 tandem duplications and 28 chromosome segmental duplications existed in the *BrAS2* genes, as shown in Figure 1A and Appendix A. The number of segmental duplications was 5.6 times the number of tandem duplications; thus, it is possible that the *BrAS2* genes could have expanded in the *B. rapa* genome through this mechanism. We analyzed the selection of gene pairs in *BrAS2* using the Ka/Ks ratio. Ka was much less than Ks (Ka/Ks < 1) for nearly all segmentally duplicated gene pairs, indicating that the replicated gene pairs had undergone strong purifying selection; the Ka/Ks ratio was highest for *BrAS2-9*/*BrAS2-44* (0.47), as shown in Appendix A, which confirmed the amplification of the *AS2* genes in two modes during evolution, namely fragment replication and tandem replication. To understand the kinship and evolutionary characteristics among the *BrAS2* family members, evolutionary trees of the *AS2* gene in *B. rapa*, *A. thaliana*, and *H. vulgare* were constructed based on sequence similarity, as described in Figure 1B. We compared the aa sequences of the AS2 proteins in *A. thaliana*, *B. rapa*, and *H. vulgare*, and constructed a phylogenetic tree of the *AS2* gene family among *A. thaliana*, *B. rapa*, and *H. vulgare,* using the neighbor-joining algorithm. We classified the *BrAS2* genes into six subgroups of Ia, Ib, Ic, Id, IIa, and IIb, which contained 23, 5, 18, 7, 7, and 2 genes, respectively. Synteny analysis is a critical analytical strategy in comparative genomics used to assess the molecular evolutionary relationships between species [34]. We conducted a collinear analysis between *B. rapa* and *A. thaliana*, as well as between *B. rapa* and *O. sativa*, as shown in Figure 1C. The findings revealed that *B. rapa* and *A. thaliana* exhibited greater homology than *B. rapa* and *O. sativa*, suggesting that the genes may have similar biological functions, providing a direction to explore the functions of *BrAS2* genes.

### 2.3. Subcellular Localization Analysis

To determine the subcellular localization of the 62 *BrAS2* genes and lay a foundation for comprehending the mechanics of gene function, we employed the PSORT online tool for subcellular localization analysis, as shown in Figure 2. Our findings revealed that the majority of the *BrAS2* genes were distributed in the nucleus, while *BrAS2-17*, *BrAS2-38*, *BrAS2-50*, and *BrAS2-24* were situated extracellularly. As plant-specific transcription factors, the *AS2* gene family may serve as a pivotal player within the nucleus.

### 2.4. Gene Structure and Conserved Motif Analysis

To analyze the similarities and differences in the *BrAS2* genes at the nucleic acid and protein levels and to speculate on the structural, functional, and evolutionary relationships among these genes, we visualized the conserved domains and scrutinized the exon-intron architecture of the *BrAS2* genes. The phylogenetic tree of the 62 *BrAS2* genes was divided into six branches, as shown in Figure 3A. Genetic structure analysis revealed that the number of exons in the *BrAS2* genes ranged from 1 to 5, as depicted in Figure 3C. We predicted 15 conserved *BrAS2* gene motifs, as described in Figure 3D. The results of the conserved motifs analysis of the BrAS2 proteins showed that motif1 and motif2 were highly conserved, with 96.8% and 93.5% of the genes containing these two motifs, respectively. In addition, motif 3 contained a leucine zipper-like structure, as revealed in Figure 3E, which facilitates specific binding with target proteins and promotes protein interactions. More intriguingly, further comparative analysis of the motifs revealed that motif7 and motif13 could contribute to the functional diversity of the *BrAS2* gene family, as they were specifically harbored in all members of the IIa subfamily, and the expression levels of many of their members varied under different treatments. The number, type, and arrangement of the *BrAS2* gene motifs located on the same branch were similar, and the functional differences in *BrAS2* genes may be due to differences in the distribution of the conserved motifs. These findings agreed with our gene structural analysis and confirmed the subfamily division of the *BrAS2* genes.

### 2.5. cis-Element Analysis

To understand the mechanisms by which *BrAS2* genes mediate the responses to abiotic stress, the 2000-bp upstream region of the *BrAS2* coding sequences was utilized to predict the *cis*-elements. Among the 62 *BrAS2* genes, 746 *cis*-acting elements were revealed, comprising 20 distinct types, as described in Figure 4, and encompassing the growth and developmental response, phytohormone response, and the stress response. All of the *BrAS2* genes contained *cis*-elements, except *BrAS2-36*, with *BrAS2-4* containing the largest (up to 25). In addition, these *cis*-elements included hormone-related elements, such as the methyl jasmonic acid (MeJA) response element and the salicylic acid response element, and stress response elements, such as cold stress, drought-induced, mechanical injury, and anaerobic-induced response elements. These sequence motifs may act as *cis*-elements, putatively participating in hormone-mediated regulation of the promoters. Notably, the light response element proved to be the most prevalent among these elements, appearing in 89% of genes, accounting for 24.1% (180) of the total. Additionally, anaerobic induction elements were discovered in 87% (54) of the genes, comprising 19.6% of the total, while MeJA response elements were identified in 69% (43) of the genes, accounting for 45% of the total number of hormone-responsive elements. These statistics indicate that *BrAS2* genes may play a vital role in abiotic stress and hormone responses.

### 2.6. GO Analysis

Annotating and conducting an enrichment analysis on the *BrAS2* genes allowed correlation with actual biological processes and specific signaling pathways. Therefore, a functional analysis was conducted using GO annotation and enrichment terms, including molecular functions (MF), cellular components (CC), and biological processes (BP). Unfortunately, the MF family was not enriched, as shown in Figure 5 and Appendix A. However, the GO-CC enrichment results identified one enriched term, specifically the nucleus (GO:0005634). The GO-BP enrichment results revealed three enriched terms of positive regulation of DNA templated transfer (GO:0045893), regulation of gene expression (GO:0010468), and hormone-mediated signaling pathways (GO:0009755). Remarkably, 11 *BrAS2s* genes were enriched in the hormone-mediated signaling pathways with a low *p*-value and high confidence. Additionally, 11 genes were enriched in the regulation of translation and DNA templated transfer, 11 genes were enriched in the nucleus, and 8 genes were enriched in the regulation of gene expression.

### 2.7. Tissue-Specific Expression Analysis

To investigate the spatial expression characteristics and potential functions of *BrAS2* genes, we examined transcriptome data from various tissues, as shown in Figure 6 and Appendix A. Our findings revealed that *BrAS2-47* exhibited a high expression level in all tissues, indicating its involvement in a wide array of developmental and regulatory processes. In contrast, *AS2-50* was exclusively expressed in siliques, suggesting its potential role in fruit development. *BrAS2-4*, *BrAS2-46*, *BrAS2-10*, and *BrAS2-39* exhibited high expression levels in roots, suggesting their potential functionality in responding to abiotic stresses, such as salt and drought. *BrAS2-8* and *BrAS2-43* displayed high expression levels in flowers, which may be related to their developmental process. Our results suggested that *BrAS2* genes not only regulate meristem development but also participate in the development of rhizomes and leaves, environmental responses, and sexual reproduction (as evidenced by the highly expressed genes in flowers).

### 2.8. Expression Patterns in Response to Abiotic Stress Analysis

Based on a transcriptome data analysis of *B. rapa* subjected to cold, salt, drought, and heat treatments, as depicted in Figure 7A–D and Appendix A, the expression levels of *BrAS2-47* and *BrAS2-10* decreased significantly under cold, salt, and drought treatments. Furthermore, the expression level of *BrAS2-56* decreased remarkably under salt stress, while that of *BrAS2-58* increased significantly under heat stress. Collectively, these findings strongly suggest that these genes play a role in counteracting abiotic stress.

### 2.9. RT-qPCR Analysis

For further screening of key genes in the *BrAS2* gene family in response to stress, we selected the most potentially functional genes from the RNA-seq analysis for RT-qPCR examination. During the cold treatment, as depicted in Figure 8A, the relative expression levels of *BrAS2-47* and *BrAS2-10* were consistently downregulated at all time points, while the expression of *BrAS2-56* increased gradually, peaking at 12 h. *BrAS2-4* expression was initially downregulated, reaching its lowest point after 4 h, and then continuously increased until 12 h. Under the drought treatment, as shown in Figure 8B, *BrAS2-47* and *BrAS2-10* exhibited identical trends, while the expression patterns of *BrAS2-56* and *BrAS2-4* decreased from 0 to 4 h and from 6 to 12 h. The expression level of *BrAS2-56* was lower at 6 h than at 0 h, while the opposite was true for *BrAS2-4*. During the salt treatment, as revealed in Figure 8C, *BrAS2-47* and *BrAS2-10* demonstrated similar trends, while the expression of *BrAS2-56* increased significantly, peaking at 4 h, and then decreased until 12 h. The *BrAS2-4* expression initially increased, then decreased, and then increased again, reaching its maximum at 12 h. Interestingly, among these four genes, *BrAS2-47* and *BrAS2-10* were downregulated in relative expression under all three stresses.

### 2.10. Prediction of Protein–Protein Interaction Network Analysis

Proteins are integral to life and interact with their surroundings to facilitate a plethora of biological processes [35]. The closeness of the evolutionary relationship between *B. rapa* and *A. thaliana* allows us to predict the function of corresponding homologous genes in *B. rapa* through protein–protein interaction (PPI) analysis of the *AS2* gene in *A. thaliana*, as the *AtAS2* gene family has been thoroughly studied; thus, the function of the *BrAS2* family was further established. Drawing upon the resources and algorithms integrated into the string database, we constructed a visual representation of the predicted *AtAS2* gene PPI network, as depicted in Figure 9A and Appendix A. Previous analyses revealed that *AT5G67420* (homologous with *BrAS2-47* and *BrAS2-10*) plays a pivotal role in regulating anthocyanin and nitrogen metabolism [21]. In addition, *AT5G67420* interacts with NIN-like protein 7 (NLP7), MYBL2, Hypersensitivity to low pi-elicited primary root shortening 1 (HRS1), and Calcium-Dependent Protein Kinase 28 (CPK28); as shown in Figure 9B, all of these have been demonstrated to be associated with the growth and development of plants, as well as their responses to abiotic stress [36,37,38,39,40,41]. *AT3G49940* (homologous with *BrAS2-58*) impedes anthocyanin synthesis and affects the extra nitrogen response [21], as revealed in Figure 9C, and in addition to interacting with NLP7, it also communicates with the BTB/POZ and TAZ domain-containing protein 2 (BT2) [42,43]. These findings further underscore the critical and multifaceted functions of *BrAS2* genes.

### 2.11. Phosphorylation Site Analysis

Protein kinases play crucial roles in regulating cellular processes by catalyzing the transfer of phosphate groups to specific amino acid residues in target proteins [44]. To identify these processes, known as protein phosphorylation in *BrAS2s*, we conducted a phosphorylation site analysis, as shown in Figure 10. All 62 members of the *BrAS2* gene family contain phosphorylation sites. We uncovered 1406 phosphorylation sites, with serine residues accounting for 65.1%, threonine residues accounting for 23.3%, and tyrosine residues accounting for 11.6%. Upon further investigation, about 89% of *BrAS2* genes simultaneously contained phospho-serine, phospho-tyrosine, and phospho-tryptophan sites. Interestingly, a significantly higher prevalence of phospho-serine sites was detected in nearly 97% of the *BrAS2* genes. Research has indicated that phosphorylated serine residues play crucial roles in cellular signal transduction, metabolic regulation, and apoptosis [45]. This finding suggests the potential biological function of this gene family.

## 3. Discussion

The *AS2* gene family is a plant-specific transcription factor family that plays an essential role in growth, the stress response, and hormone induction in many plants. However, there have been no relevant reports on *BrAS2* genes. Here, we identified 62 *AS2* genes in *B. rapa* and analyzed their physicochemical properties and protein structures, as well as their expression patterns in response to different abiotic stresses. Our results improve the understanding of *BrAS2* genes and the role they play in the response to abiotic stress in plants.

We considered the evolution of the *BrAS2* family genes. We revealed that the *BrAS2* gene duplicated 28 chromosomal segments, as a 5.6 times tandem duplication, indicating that segment duplication may be the key factor for expansion of the *BrAS2* gene in the *B. rapa* genome. In addition, the Ka/Ks ratio of duplicated gene pairs in *BrAS2s* was significantly less than 1, indicating that *BrAS2* genes underwent purifying selection. Previous studies have revealed that *Brassicaceae* underwent three whole genome replication events [46], resulting in the acquisition of highly intricate gene families. The number of *BrAS2* genes (*N* = 62) superseded that of *AtAS2* genes (*N* = 35), which was attributed to the unique whole-genome tripling event (WGT) in *Brassica* [47,48]. Furthermore, one *AtAS2* gene was mapped to as many as five *BrAS2* genes, reflecting the WGT event in *Brassica*. In addition, a collinearity analysis of the *BrAS2* genes with *AtAS2* genes and *OsAS2* genes indicated that the number of collinearity gene pairs between the *BrAS2* genes and the *AtAS2* genes was more than 16-times higher than that of the *BrAS2* genes and the *OsAS2* genes. This points to a closer evolutionary relationship between the *BrAS2* genes and the *AtAS2* genes compared to the *BrAS2* genes and the *OsAS2* genes. Therefore, we inferred the function of the *BrAS2* genes based on the function of the *AtAS2* genes. 

A whole-genome replication event can cause the diversification of structures and functional domains. The *AS2* gene family has been categorized into two types based on the structure of the LOB domain [1,2]. Class I *LBD* genes feature a completely conserved CX2CX6CX3C zinc finger-like domain (motif2) and an LX6LX3LX6L leucine zipper (motif3), while class II *LBD* genes only contain a conserved zinc finger-like domain [1]. These protein domains play a pivotal role in various aspects of plant growth and development, as well as in defending against external stresses. Class I *LBD* genes primarily participate in development, namely the formation of lateral organs, such as leaves and flowers [14,49], as well as the transduction cascade of auxin signals [14,50,51], which leads to the building of lateral roots. In contrast, class II genes are engaged in metabolism, particularly as suppressors of anthocyanin synthesis and N availability signals in plants [52,53].

Mature proteins are transported inside specific organelles to perform stable biological functions. Among the subcellular localization results, most of the BrAS2 proteins were predicted to be located in the nucleus and a GO analysis of the *BrAS2* genes showed that the BP of enrichment was mainly related to hormone-mediated signaling and positive regulation of transcription. In addition, several *cis*-elements were related to environmental stress. We speculate that *BrAS2* genes regulate the transcription of relevant genes and adapt to the environment by sensing environmental stress through hormone signaling pathways.

Related studies have shown that MeJA improves drought tolerance in *O. sativa* [54]. In this study, among the *cis*-elements associated with the hormone responses, the largest proportion was MeJA-responsiveness, at 45%. In addition, salt stress response mechanisms were also regulated by MeJA [55]. For instance, MeJA antagonizes the adverse effects of osmotic stress by regulating inorganic penetrating ions or organic penetrants to suppress the absorption of toxic ions [56]. Furthermore, MeJA participates in the resistance to cold stress. MeJA plays a crucial role in the *S. lycopersicum* response to cold stress by promoting ABA biosynthesis [57]. Interestingly, studies reported that an exogenous application of MeJA improves heat tolerance in *Lolium perenne* by mediating the expression of genes in different pathways, such as chlorophyll biosynthesis and degradation, antioxidant enzyme systems, the HSF-HSP network, and JA biosynthesis [58]. In combination with the *BrAS2* gene expression transcriptome data under drought, salt, and cold stress, the expression levels of *BrAS2-10* and *BrAS2-47* were significantly downregulated, while *BrAS2-58* expression was significantly upregulated under heat stress. *BrAS2-47* predicted more MeJA-responsive *cis*-elements, so there was a higher possibility that it was involved in the regulatory role of the MeJA response to stress.

Protein phosphorylation is a fundamental and ubiquitous signal transduction mechanism that plays a pivotal role in the response to various abiotic stresses [59]. Protein phosphorylation predominantly occurs on threonine, serine, and tyrosine residues [60]. By investigating the phosphorylation sites of the *BrAS2* family of proteins, we revealed 1406 phosphorylation sites, with serine residues accounting for 65.1%. We inferred that the biological function of this gene family was likely determined by the phosphorylation of serine residues, and serine plays a pivotal role in cellular signal transduction, metabolic regulation, and more, providing evidence for the functionality of the *BrAS2* gene family [45]. In terms of tissue-specific expression, *BrAS2-47* exhibits high expression levels across all tissues, while *BrAS2-4*, *BrAS2-10*, *BrAS2-56,* and *BrAS2-58* demonstrate high expression levels in roots. *BrAS2-8*, on the other hand, displays high gene expression in leaves. We postulated that these genes might play a crucial role in mitigating environmental stresses.

The perception of adverse environmental conditions triggers a stress-specific signaling cascade. This complex process requires the interaction of second messengers, including Ca^2+^, reactive oxygen species (ROS), nitric oxide, and phospholipids, as well as post-translational modification of proteins [59]. In plants, signal transduction in response to abiotic stresses is characterized by ROS signal transduction, calcium signal transduction, and protein phosphorylation [61]. We analyzed the expression profiles of *BrAS2* gene family genes under different stresses. According to the transcriptome analysis, as revealed in Figure 7, *BrAS2-4* was significantly downregulated under the salt and drought treatments, while *BrAS2-10* was downregulated under the low-temperature, salt, and drought treatments. *BrAS2-47* was downregulated to varying degrees under the different stress treatments, and *BrAS2-56* was downregulated under the salt treatment. In contrast, *BrAS2-58* expression was significantly upregulated under the high-temperature treatment. We used RT-qPCR to further validate the transcriptome results. The RT-qPCR analysis revealed that the expression levels of *BrAS2-47* and *BrAS2-10* were substantially downregulated under cold stress, whereas that of *BrAS2-56* was upregulated initially and then downregulated. Furthermore, the expression levels of *BrAS2-47*, *BrAS2-10*, and *BrAS2-56* were downregulated under drought stress, while the expression of *BrAS2-4* was initially decreased, followed by an increase, and then another decline. Under the salt stresses, *BrAS2-47* and *BrAS2-10* exhibited a similar trend to the other treatments, whereas *BrAS2-4* increased and *BrAS2-56* initially rose and then declined. We inferred from these results that *BrAS2-47* and *BrAS2-10* were responsive to cold, drought, and salt stress, and contained promoters, such as drought-inducibility and anaerobic induction. *BrAS2-58* was specifically induced by heat and contained elements related to defense and stress responsiveness. Interestingly, hormone response elements, such as MeJA, salicylic acid, and gibberellin responsiveness elements, were also widely distributed among these three genes. Furthermore, the expression levels of the homologous genes in *A. thaliana* varied greatly under the different stress treatments, and both genes positively responded to ABA stress, as revealed in Appendix A, which further supports the proposed function of these three genes. Therefore, our preliminary hypothesis was that these three genes play a vital role in the abiotic stress response through hormone signaling pathways: *BrAS2-10* and *BrAS2-47* responded to salt, cold, and drought stress through negative regulation, while *BrAS2-58* was responsive to heat stress through positive regulation.

We discovered that these two genes interacted with NLP7 and MYBL2 through our prediction of the PPI network between *AT5G67420* (homologous genes *BrAS2-47* and *BrAS2-10*) and *AT3G49940* (homologous gene *BrAS2-58*). Upon detection of nitrate, the PB1 domains of the *NLP* transcription factors orchestrated gene expression from chromosomal DNA via homo- and hetero-oligomerization in the presence of nitrate, culminating in the modulation of gene expression to promote nitrate uptake and utilization, thereby conferring resilience against abiotic stresses such as drought and salinity [36,37]. *MYBL2* inhibits the biosynthesis of anthocyanins in *A. thaliana* through mediation by ABA [38]. In addition, *AT5G67420* also interacts with HRS1 and CPK28. The pivotal role of HRS1 proteins in coordinating nitrogen and phosphorus absorption and utilization in response to abiotic stress has been demonstrated [40]. CPK28 was phosphorylated and promoted nuclear translocation of the NLP7 protein, thus specifying the transcriptional reprogramming of cold-responsive gene sets in response to Ca^2+^ [41]. Additionally, *AT5G67420* was confirmed to regulate anthocyanin and nitrogen metabolism, thereby further supporting this conclusion [21]. We speculated that *AT5G67420* interacted with the NLP7, MYBL1, CPK28, and HRS1, thereby affecting the resistance to salt, drought, and low-temperature stress. In addition to its interaction with NLP7 and MYBL1, *AT3G49940* interacted with BT2 and MYB11. The functional deletion mutant analysis of BT2 verified that both *BT2* transcription factors functioned in an ABA-dependent manner, to regulate germination and development during sugar signal transduction [43]. Furthermore, MYB11 produced flavonol glycoside 2, which reduces the proliferation activity of meristem cells and delays development in response to heat stress [43]. We speculate that *AT3G49940* could enhance the yield of Chinese cabbage in light of the current climate and environmental changes by interacting with NLP7, MYBL1, BT2, and MYB11.

## 4. Materials and Methods

### 4.1. Identification of AS2 Family Gene Members in B. rapa

Whole-genome sequences, gff3 genome annotation data, and AS2 aa sequences of *B. rapa, A. thaliana*, and *H. vulgare* were downloaded from EnsemblPlant (http://plants.ensembl.org/, accessed on 1 June 2023). The *BrAS2* genes were screened from the *B. rapa* genome using the BLASTP program with AtAS2 aa sequences as input to predict the *BrAS2* genes. The online tool NCBI Batch CD Search was used to analyze the aa sequence of the *AS2* genes protein in *B. rapa*, and remove the protein sequence that did not contain the AS2 domain.

### 4.2. Chromosomal Localization and Collinearity Analysis

The location of the *BrAS2* genes on the chromosomes was extracted from the *B. rapa* gff3 genome annotation information using TBtools (v1.120) [62]. The MCScanX plug-in in TBtools (v1.120) was used to analyze the relationships among duplicate genes within *B. rapa* and inter-specific collinearity, and the Circos plug-in was used to visualize the results.

### 4.3. Subcellular Localization Analysis

The isoelectric points and molecular weights of the aa sequences of all *B. rapa* AS2 proteins were predicted using the ExPASy technologies server (http://www.expasy.org/, accessed on 1 June 2023) [63], and the subcellular distribution was analyzed using PSORT.

### 4.4. Phylogenetic Tree Construction, Conserved Motifs, and Gene Structural Analysis

The aa sequences of *BrAS2s, AtAS2s,* and *HvAS2s* were aligned using the MUSCLE algorithm in MEGA X [64], and an unrooted phylogenetic tree was constructed using the maximum likelihood method in MEGA X with 1000 bootstrap replicates. The aa sequences of the *BrAS2* genes were uploaded to the MEME online website (http://meme-suite.org/, accessed on 1 June 2023), and the number of motifs was set to 15 to analyze the conserved *BrAS2* gene motifs. The gene structure, motifs, and conserved domains were visualized using TBtools (v1.120) [62].

### 4.5. Cis-Element and Gene Ontology Analyses

The upstream 2000-bp sequence of each *BrAS2s* was analyzed using the online PlantCARE database (http://bioinformatics.psb.ugent.be/webtools/plantcare/html/, accessed on 1 June 2023) [65], with default parameters. The *BrAS2* gene structure and conserved motifs were visualized using TBtools (v1.120). A gene ontology (GO) analysis of the *BrAS2* genes was conducted with default parameters at the DAVID website (http://david.ncifcrf.gov, accessed on 1 June 2023) [66].

### 4.6. Plant Material, Stress Treatments, and Total RNA Extraction

*B. rapa* with stable self-incompatibility was used for the stress treatments. Plump seeds were seeded in MS Modified Medium (with vitamins, Sucrose, Agar) (PM10121-307, Coolaber, Beijing, China) and cultivated in a plant incubator. Seedlings with six leaves and similar growth status were subjected to the stress treatments. The seedlings were placed in a hydroponic system with 150 mM NaCl to simulate salt stress and in 15% PEG6000 to simulate drought conditions. The plants were exposed to 4 °C for the cold stress treatment, and exposed 16 h 45 °C/8 h 35 °C for the heat stress treatment. We used unstressed *B. rapa* seedlings with the same growth period and under the same growth conditions as a control (CK). The duration of all stress treatments was 4, 6, and 12 h. Three biological replicates were run for each treatment group, and the samples were stored at −80 °C. Total RNA was extracted using a FastPure^®^ Cell/Tissue Total RNA Isolation Kit V2 (Vazyme Biotech Co., Ltd., Nanjing, China).

### 4.7. Expression Patterns Analysis

The transcriptome sequences in different *B. rapa* tissues from NCBI GEO with Accession number GSE43245 were downloaded, and the data were normalized using the TPM method. The transcriptome sequences of *A. thaliana* under various abiotic stress treatments were downloaded, as shown in Appendix A. Gene expression profile heatmaps were prepared using TBtools (v1.120) [62].

The samples from the treatments were subjected to transcriptome sequencing on the Illumina NovaSeq 6000 platform by BioMarker Technologies (Beijing, China), and three biological replicates were collected for each sample. The number of mapped reads and transcript lengths in the samples were normalized after sequencing. The abundance of the transcripts was measured using fragments per kilobase of transcript per million fragments mapped.

### 4.8. RT-qPCR Analysis

RNA samples were reverse-transcribed using TransScript^®^ Uni All-in-One First-Strand cDNA Synthesis SuperMix for the RT-qPCR analysis. RT-qPCR was performed on a qTOWER3 qPCR machine using TransStart^®^ Green qPCR SuperMix (TransGen Biotech, Beijing, China), and *BrACTIN2* was used as the reference gene. The primer sequences are shown in Appendix A.

### 4.9. Statistical Analysis

The relative expression levels of each gene were analyzed using the 2^−ΔΔCt^ method, and the analysis of significant differences (a, b, c, d) was conducted through the implementation of single-factor ANOVA test on IBM SPSS Statistics 25, in order to compare the obtained means (with a = 0.05).

### 4.10. Protein–Protein Interaction Networks and Phosphorylation Site Analysis

The STRING online website was used to predict the PPI relationships with default parameters, and Cytoscape v3.9.1 was used to construct the interaction network [67]. The *BrAS2* gene phosphorylation sites were analyzed using the online Netphos 2.0 software Server, and Excel 2016 was used to prepare the visual representations.

## 5. Conclusions

In this study, 62 *BrAS2s* genes were identified from the *B. rapa* genome. After a comprehensive analysis of the sequence features, the expression profiles, and the protein interactive relationships, we determined that *BrAS2-10* and *BrAS2-47* had the greatest potential in regulating cold, salt, and drought tolerance, and *BrAS2-58* was involved in the *B. rapa* high-temperature response.

## Figures and Tables

**Figure 1 ijms-24-10534-f001:**
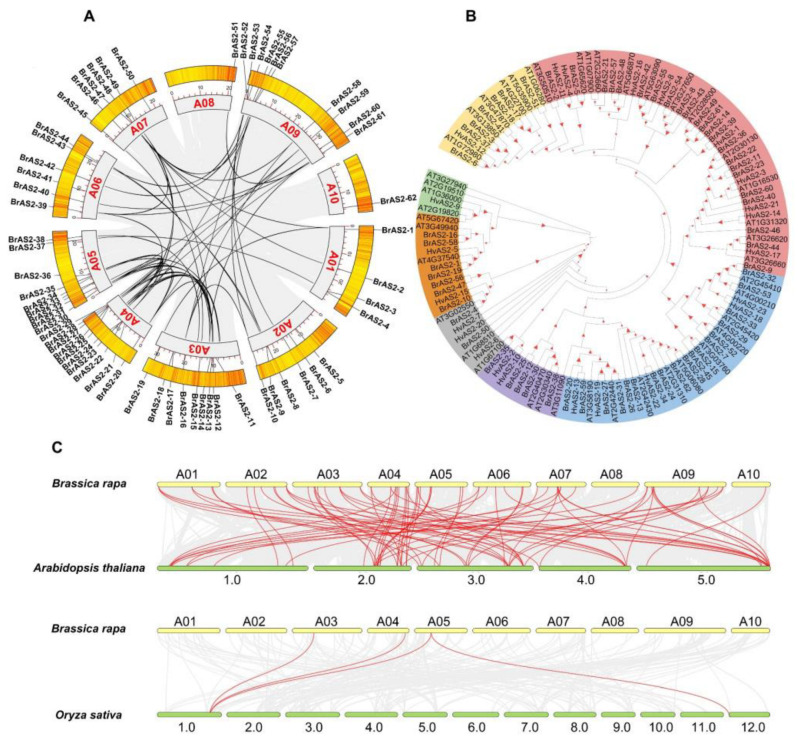
(**A**) Chromosome location and replication of *AS2* Gene Family members in *Brassica rapa*. (**B**) The phylogenic tree of *B. rapa, Arabidopsis thaliana,* and *Hordeum vulgare*. The clustering analysis was based on 1000 replications to increase the credibility of the bootstrap value. Red characters represent Class Ia; purple characters represent Class Ib; blue characters represent Class Ic; yellow characters represent Class Id; green charactes represent Class Ie; brown characters represent Class IIa; and grey characters represent Class IIb. (**C**) Collinearity among *AS2* Gene Family between *B. rapa* and *A. thaliana, B. rapa* and *Oryza sativa*. The red line represents the homologous genes between *B. rapa* and *A. thaliana*, and between *B. rapa* and *O. sativa*.

**Figure 2 ijms-24-10534-f002:**
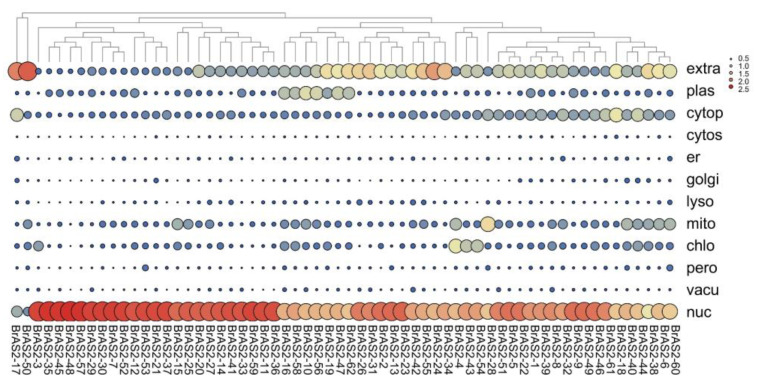
The prediction of subcellular localization for *AS2* genes in *B. rapa*. The color and the size of the circle indicated the values of the reliable index of the prediction results. TMHMM2.0 predicted that *BrAS2-24* had a transmembrane structure of 173–191 amino acids.

**Figure 3 ijms-24-10534-f003:**
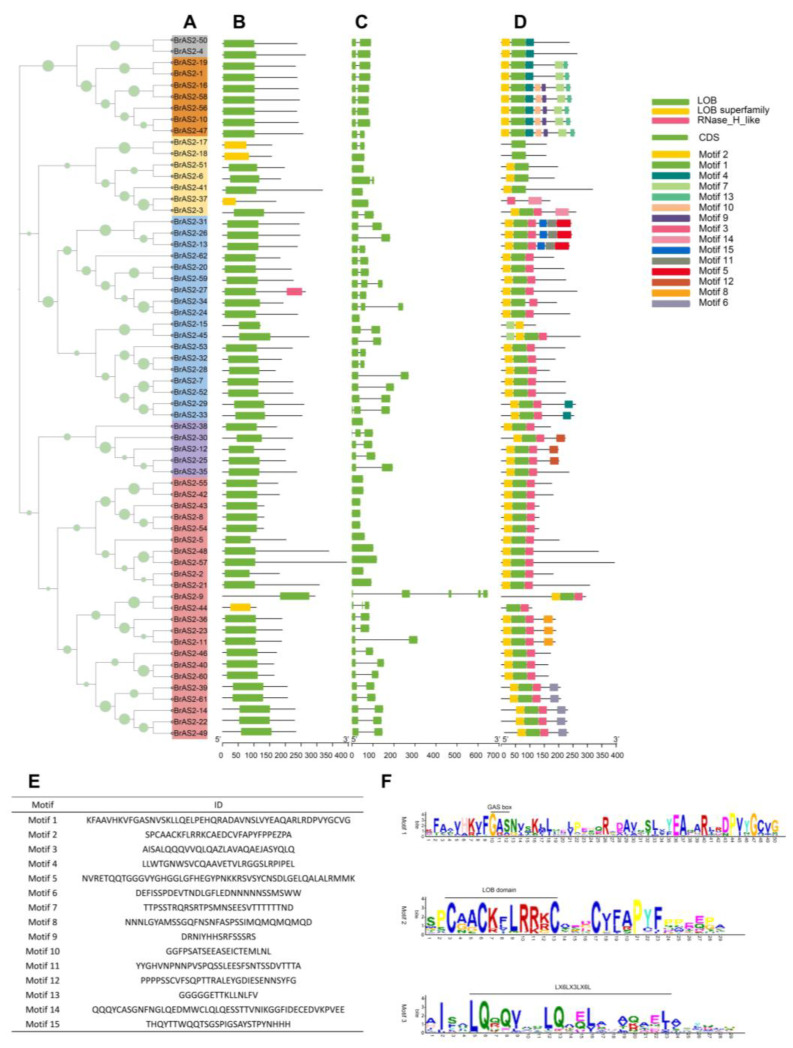
(**A**) Phylogenetic relationships of the *BrAS2* gene family. (**B**) The distribution of the conserved domain. (**C**) The gene structure. (**D**) The distribution of the conserved motif. (**E**) The sequences of 15 Motifs in the *BrAS2* gene family. (**F**) Three symbolic conserved motif logos.

**Figure 4 ijms-24-10534-f004:**
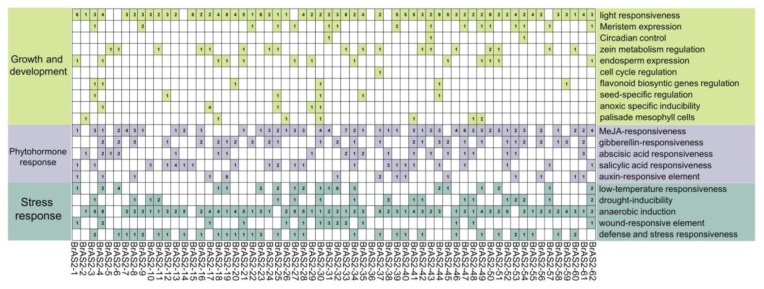
*cis*-elements of 2000 bp sequences upstream of *BrAS2* genes. The statistics of *cis*-elements for each *BrAS2* gene. Green, purple, and teal signify the presence of a specific motif, with numbers denoting the quantity. Conversely, white denotes the absence of a unique motif. The various hues flanking the squares indicate distinct biological processes.

**Figure 5 ijms-24-10534-f005:**
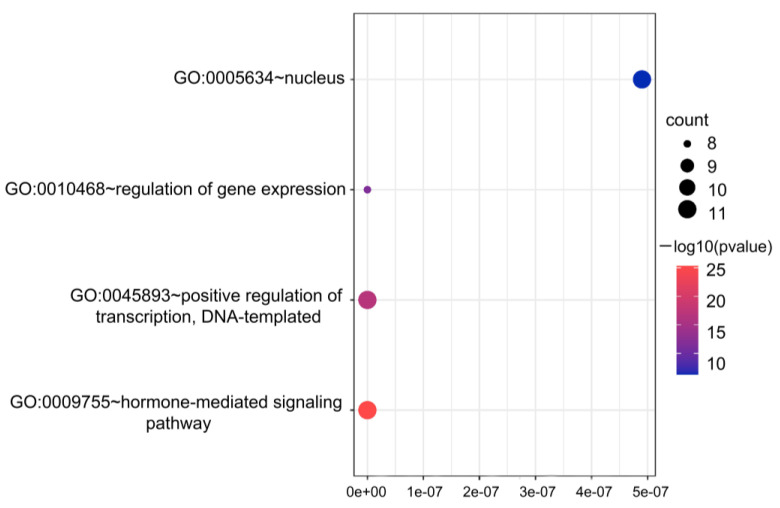
GO enrichment analysis of *BrAS2s*. The size of the dot bubble represents the number, and the color represents the *p*-value of genes for that GO term.

**Figure 6 ijms-24-10534-f006:**
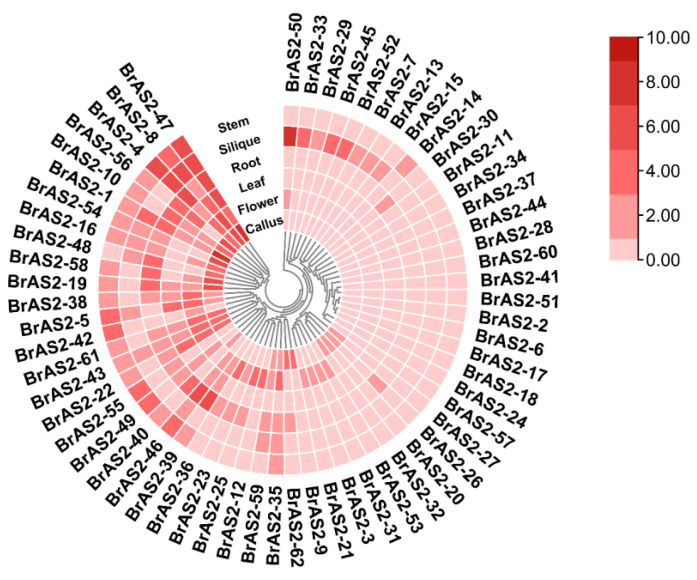
Tissue-specific expression of *BrAS2* genes. The transcriptome sequences in different *B. rapa* tissues from NCBI GEO with the accession number GSE43245 were downloaded, and the data were normalized using the TPM method. The heatmap shows the *BrAS2* gene expression level across six tissues, including the stem, silique, root, leaf, flower, and callus. In this figure, a continuous gradient of a single red color represents the different gene expression levels in different tissues, with deeper shades of red indicating higher expression levels and, conversely, lighter shades indicating lower expression levels.

**Figure 7 ijms-24-10534-f007:**
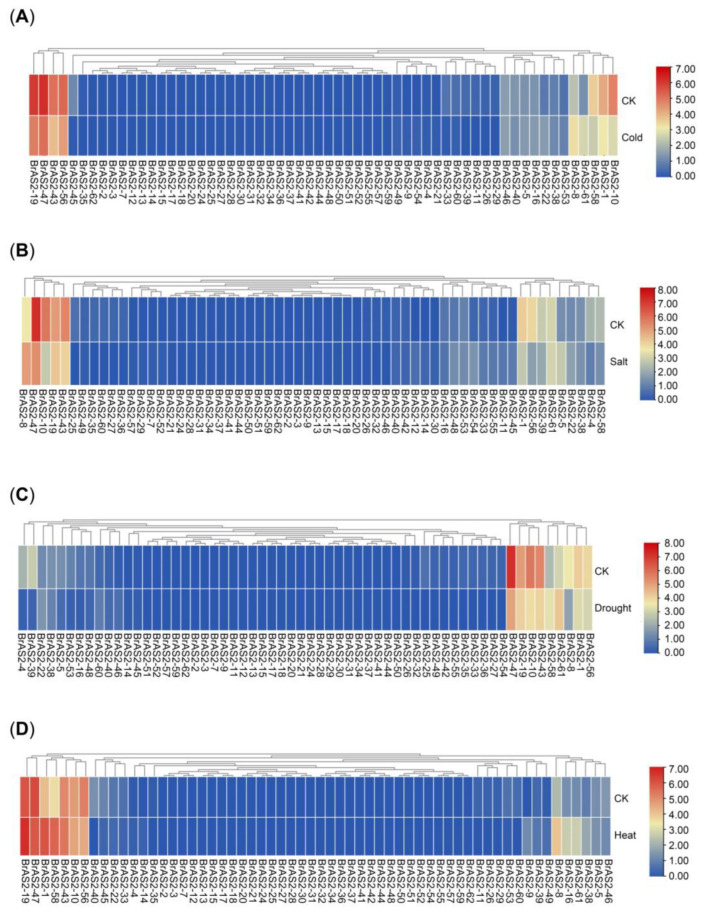
Heatmaps of the expression profile of *BrAS2* under several different abiotic stresses. (**A**) Under cold. (**B**) Under salt. (**C**) Under drought. (**D**) Under heat. We used unstressed *B. rapa* seedlings at the same growth period and under the same growth conditions as CK.

**Figure 8 ijms-24-10534-f008:**
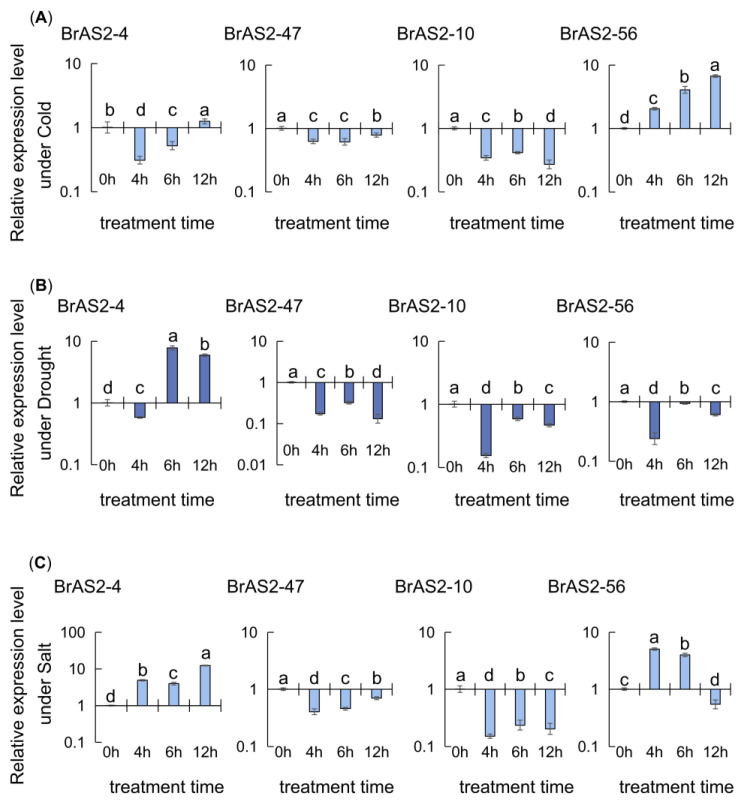
Expression profiles of 4 *BrAS2* genes under abiotic stress were analyzed using RT-qPCR. (**A**) Under cold. (**B**) Under drought. (**C**) Under salt. The above experiments were performed using 0 h as the control (CK), and the treatment time was set to 4, 6, and 12 h. Each group was subjected to three biological replicates, and error bars indicate standard errors. All values were logarithmized. Letters above data bars indicate the statistical significance (the means are arranged in descending order, with the letter “a” after the highest mean, a = 0.05).

**Figure 9 ijms-24-10534-f009:**
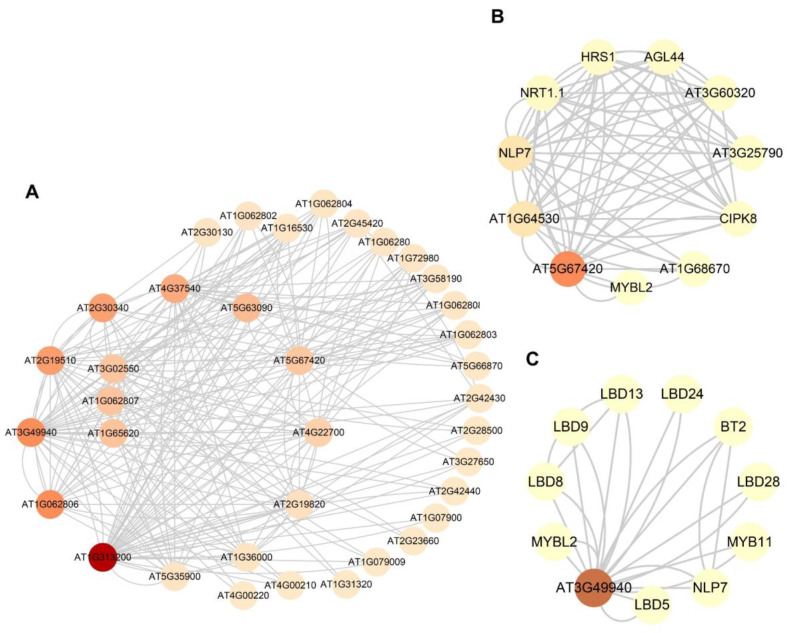
(**A**) Protein–protein interaction networks of AS2 protein in *A. thaliana*. (**B**) *AT5G67420* (*BrAS2-47* and *BrAS2-10* homologous gene) PPIs. (**C**) *AT3G49940* (*BrAS2-58* homologous gene) PPIs. Each node represents a protein, and the lines between nodes represent the interactions between proteins. Node size and fill color shade are positively correlated with degree centrality.

**Figure 10 ijms-24-10534-f010:**
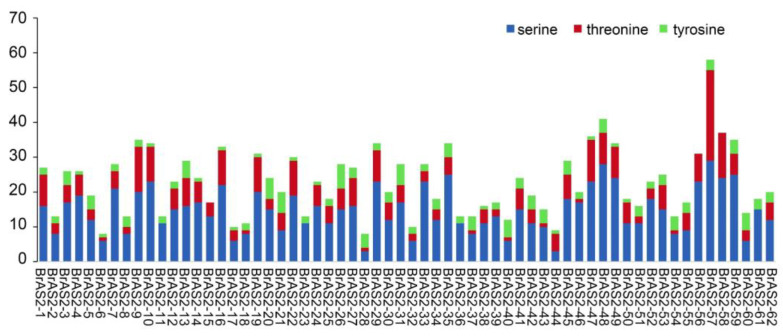
Distribution of predicted phosphorylation sites in the amino acid sequence of BrAS2s.

**Table 1 ijms-24-10534-t001:** Information on the *AS2* family in *Brassica rapa*.

Gene Name	Gene ID	Chromosome Distribution	Classification	Protein Length (aa)	M_W_ (Da)	PI	*A. thaliana* ID
*BrAS2-1*	*Bra011772*	A01 (631422–632295)	IIa	237	26,167.73	8.14	*AT4G37540*
*BrAS2-2*	*Bra030013*	A01 (15744550–15745095)	Ia	181	20,472.25	6.28	*AT3G50510*
*BrAS2-3*	*Bra021513*	A01 (24855421–24856203)	Id	260	29,654.86	5.19	*AT3G13850*
*BrAS2-4*	*Bra021433*	A01 (25233613–25234491)	IIb	264	28,181.17	8.9	*AT3G02550*
*BrAS2-5*	*Bra039733*	A02 (9622401–9623009)	Ia	202	22,093.85	7.1	*AT1G65620*
*BrAS2-6*	*Bra008062*	A02 (12966840–12967397)	Id	185	21,098.41	8.3	*AT1G72980*
*BrAS2-7*	*Bra008514*	A02 (16097898–16100589)	Ic	224	23,732.84	6.53	*AT4G00220*
*BrAS2-8*	*Bra033019*	A02 (21869558–21869956)	Ia	132	14,678.52	6.89	*AT3G27650*
*BrAS2-9*	*Bra032938*	A02 (22395463–22401919)	Ia	294	32,145.26	4.78	*AT3G26660*
*BrAS2-10*	*Bra031833*	A02 (27641733–27642526)	IIa	241	25,985.18	8.46	*AT5G67420*
*BrAS2-11*	*Bra022780*	A03 (7018824–7021941)	Ia	188	20,973.78	6.21	*AT2G30130*
*BrAS2-12*	*Bra000188*	A03 (9790598–9791563)	Ib	199	21,455.45	8.8	*AT2G40470*
*BrAS2-13*	*Bra000257*	A03 (10204676–10206488)	Ic	238	25,950.97	6.88	*AT2G42430*
*BrAS2-14*	*Bra000491*	A03 (11383294–11384774)	Ia	231	25,396.57	4.9	*AT2G28500*
*BrAS2-15*	*Bra001087*	A03 (14672898–14673260)	Ic	120	12,555.11	10.63	*AT3G03760*
*BrAS2-16*	*Bra012913*	A03 (21540949–21541767)	IIa	241	26,607.1	6.89	*AT3G49940*
*BrAS2-17*	*Bra019365*	A03 (24669629–24670232)	Id	157	17,587.86	4.73	*AT5G35900*
*BrAS2-18*	*Bra019364*	A03 (24677058–24677660)	Id	156	17,566.68	4.72	*AT5G63090*
*BrAS2-19*	*Bra017831*	A03 (30772548–30773423)	IIa	232	25,301.1	9.1	*AT4G37540*
*BrAS2-20*	*Bra014581*	A04 (1568864–1569640)	Ic	219	24,375.97	6.28	*AT3G58190*
*BrAS2-21*	*Bra032153*	A04 (10493828–10494754)	Ia	308	34,410.77	4.83	*AT2G23660*
*BrAS2-22*	*Bra035698*	A04 (12731349–12732755)	Ia	229	25,158.4	4.82	*AT2G28500*
*BrAS2-23*	*Bra021612*	A04 (13416815–13417633)	Ia	190	21,127.98	5.91	*AT2G30130*
*BrAS2-24*	*Bra021737*	A04 (14176806–14179238)	Ic	239	26,536.2	7.05	*AT2G31310*
*BrAS2-25*	*Bra016992*	A04 (17230175–17231282)	Ib	201	21,738.62	7.68	*AT2G40470*
*BrAS2-26*	*Bra016877*	A04 (17780078–17781492)	Ic	246	26,618.56	8.11	*AT2G42430*
*BrAS2-27*	*Bra016876*	A04 (17786570–17788024)	Ic	264	29,442.97	6.29	*AT2G42440*
*BrAS2-28*	*Bra040312*	A04 (18539194–18539789)	Ic	168	18,604.4	7.68	*AT2G45410*
*BrAS2-29*	*Bra040311*	A04 (18545871–18547692)	Ic	259	27,132.39	8.19	*AT2G45420*
*BrAS2-30*	*Bra004572*	A05 (848386–849380)	Ib	223	24,453.85	8.73	*AT2G40470*
*BrAS2-31*	*Bra004693*	A05 (1378014–1379047)	Ic	244	26,262.19	8.15	*AT2G42430*
*BrAS2-32*	*Bra004908*	A05 (2423168–2423825)	Ic	188	20,738.62	6.29	*AT2G45410*
*BrAS2-33*	*Bra004910*	A05 (2431903–2433701)	Ic	253	26,065.29	8.51	*AT2G45420*
*BrAS2-34*	*Bra018260*	A05 (6977723–6978406)	Ic	193	21,602.68	5.93	*AT2G31310*
*BrAS2-35*	*Bra018320*	A05 (7426670–7428607)	Ib	236	25,340.52	8.88	*AT2G30340*
*BrAS2-36*	*Bra018335*	A05 (7526963–7527792)	Ia	189	21,079	6.03	*AT2G30130*
*BrAS2-37*	*Bra027392*	A05 (20677617–20678129)	Id	170	19,756.71	4.5	*AT3G13850*
*BrAS2-38*	*Bra034867*	A05 (21803639–21804157)	Ib	172	18,765.48	6.58	*AT3G11090*
*BrAS2-39*	*Bra018675*	A06 (2680223–2681288)	Ia	206	22,866.92	4.96	*AT2G28500*
*BrAS2-40*	*Bra026042*	A06 (6372131–6373653)	Ia	163	18,075.77	8.53	*AT1G16530*
*BrAS2-41*	*Bra018102*	A06 (10251078–10252128)	Id	318	36,165.12	5.07	*AT3G47870*
*BrAS2-42*	*Bra038606*	A06 (14136819–14137364)	Ia	181	19,910.5	8.29	*AT5G63090*
*BrAS2-43*	*Bra025294*	A06 (22339538–22339936)	Ia	132	14,563.43	7.58	*AT3G27650*
*BrAS2-44*	*Bra025217*	A06 (22711432–22712256)	Ia	107	11,876.75	6.57	*AT3G26620*
*BrAS2-45*	*Bra036436*	A07 (499213–500551)	Ic	275	29,136.32	7.27	*AT3G03760*
*BrAS2-46*	*Bra014907*	A07 (5302258–5303262)	Ia	172	18,669.09	7.61	*AT1G31320*
*BrAS2-47*	*Bra012164*	A07 (9636805–9637668)	IIa	256	27,868.24	8.4	*AT5G67420*
*BrAS2-48*	*Bra012112*	A07 (9939966–9940982)	Ia	338	37,528.62	6.54	*AT5G66870*
*BrAS2-49*	*Bra011942*	A07 (11079984–11081436)	Ia	233	25,579.89	4.88	*AT2G28500*
*BrAS2-50*	*Bra004315*	A07 (17939314–17940220)	IIb	237	25,763.1	8.05	*AT1G68510*
*BrAS2-51*	*Bra030647*	A08 (20962015–20962608)	Id	197	22,476.04	9.68	*AT1G06280*
*BrAS2-52*	*Bra037323*	A09 (1067402–1069396)	Ic	224	23,588.82	6.43	*AT4G00220*
*BrAS2-53*	*Bra037322*	A09 (1073182–1074567)	Ic	222	24,234.34	6.16	*AT4G00210*
*BrAS2-54*	*Bra039072*	A09 (1261020–1261415)	Ia	131	14,582.52	7.6	*AT3G27650*
*BrAS2-55*	*Bra035860*	A09 (3268177–3268707)	Ia	176	19,265.76	8.56	*AT5G63090*
*BrAS2-56*	*Bra037847*	A09 (3873152–3873956)	IIa	236	25,719.08	9.03	*AT5G67420*
*BrAS2-57*	*Bra037142*	A09 (4287000–4288184)	Ia	394	44,228.21	5.62	*AT5G66870*
*BrAS2-58*	*Bra036040*	A09 (24873579–24874625)	IIa	245	26,699.27	6.82	*AT3G49940*
*BrAS2-59*	*Bra007385*	A09 (28936037–28936836)	Ic	225	24,753.19	6.01	*AT3G58190*
*BrAS2-60*	*Bra026716*	A09 (33360480–33361740)	Ia	164	18,506.26	9.15	*AT1G16530*
*BrAS2-61*	*Bra031599*	A09 (35646044–35647161)	Ia	207	23,134.33	4.87	*AT2G28500*
*BrAS2-62*	*Bra009161*	A10 (15474429–15475061)	Ic	183	20,624.28	5.72	*AT5G06080*

“M_W_” is molecular weight. “PI” is isoelectric point.

## Data Availability

All the data that support the findings of this study are available in the paper and its Appendix A published online.

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
