# Peer review of "Genome-Wide Identification and Expression Analysis of AS2 Genes in Brassica rapa Reveal Their Potential Roles in Abiotic Stress"

_ijms, 2023, doi:10.3390/ijms241310534_

Round 1

Reviewer 1 Report

This is a very interesting and comprehensive work, characterizing the family of BrAS2 genes.

I do have some concerns, though, especially some information that is missing from methods and/or result presentation. There are also some minor mistakes along the text.

Point by point review:

In the abstract  ‘Arabidopsis thaliana’ should be in italic.

Along the text: you don’t have to put the species abbreviated name together with the one written in full. You just have to write in full at first mention and then use the abbreviated version. Also, some species have the common name together with the Latin one (tomato, rice, mayze) but others as relevant don’t have (eg: potato, barley).

Line 35: a plant-specific transcription factor family that

Line 46-47: grapevine and soybean are not Latin names and therefore they shouldn’t be written in italic. However, as all the other species names are written in Latin, maybe you should write these two species in Latin too.

58-59 growth response to water deficit

83 studies in Brassica campestris

87 Brassica rapa (Chinese cabbage)

Line 93: if “2” is Results, everything under it should be “2…” not 1.

In “Results” overall: you have a lot of discussion in this chapter that should be moved to “discussion”, here you should just describe your findings, one example is subchapter 1.9 (that should be 2.9), that is almost completely discussion.

95 characteristics. Also, here first you identified, so the beginning of the sentence should be “To identify and understand the characteristics of the AS2 gene…”

100 remove “meticulously”

147-148 phylogenetic relationships between

151 “thus, it is possible that the BrAS2 genes could have expanded in the B. rapa genome through this mechanism”

Line 280, legend to fig 6 remove “covering the entire B. rapa life cycle,”

Legend to fig 7 does not correspond do the actual figure, especially 7C, I can’t find DT-PEG and DS-PEG in the figure ? Must define CK.

Line 383: predicted instead of predicated

Line 386: transcription factor family

Line 436: why is perennial ryegrass in italic?

The last paragraph of discussion should be in conclusions.

Materials & methods:

Sub chapter 4.6: You don´t describe the heat treatment.

Statistical analysis?

Materials & methods and presentation of results are very unclear regarding the following:

You performed RNAseq and then qPCR on what samples? Cold, drought and salt stress?

Where are your RNAseq results? qPCR is in figure 8, right?

What is shown on figs 6 and 7? Results retrieved from databases? It should be clearly stated in Materials and methods.

Namely figure 6: what is callus? Is it in vitro grown undifferentiated tissue? Then you shouldn’t use it here nor draw results from it. What is the control? The expression levels are being compared to what? You say the data were logaritmized but your downregulations are shown from 0 to 1, so there was not logaritmization, otherwise downregulation would be negative….

Figure 7: I don’t understand if these are your data or not, as you performed some of these stresses but it seems you did not perform heat stress…

Figure 8: Up and down regulation would be better seen if you logaritmized these results… I don’t understand the change of color of the bars?

Expression of BrAS2-47: in figure 6 it is clearly upregulated in all tissues, so it is very important to understand in relation to what (what was the control?), while in response to stress it is down regulated. Also, in lines 263-265 you state that it plays important roles in response to environmental stress due to its ip regulation when if fact that role is through down regulation, so remove the “environmental response” from here, and discuss its dual role in the discussion.

please see above, in "point by point" revision.

Reviewer 2 Report

In the manuscript "Genome-wide identification and expression analysis of AS2 genes in Brassica rapa reveal their potential roles in abiotic stress," the authors identified 62 BrAS2s genes from the B. rapa genome. They conclude, based on a comprehensive analysis of the sequence features, the expression profiles, and the interactive protein relationships, that the family member BrAS2-10 and BrAS2-47 have the most significant potential to regulate cold, salt, and drought tolerance, and the BrAS2-58 gene is involved in the reaction of B. rapa at high temperature. The paper addresses significant issues, uses adequate methods, and applies appropriate analyses. It is well structured, and the discussion reflects the results well.

Some notes:

Line 25: Arabidopsis thaliana must be in italics!

Line 69: In A. thaliana, AtLBD37, AtLBD38, and AtLBD39 genes.....
Line 83: Similarly, studies on in Brassica campestris....
Please, delete in!

You wrote on line 323: The letter above the data bars indicates statistical significance (a= 0.05). Please explain what mean letters: a, b, c, and d.

qRT-PCR analysis must be RT-qPCR analysis

Figure 3E: The 15th motif needs to be completely visible, and what does ID mean?

Remove the dot after 4℃ on line 558!

Line 590 B rapa must be B. rapa

Line 587: In this study, 62 BrAS2s genes.....

Round 2

Reviewer 1 Report

The authors made quite a good job in revising the MS, congratulations! They reponded to all my conserns. I only have two minor suggestions left:

513-514: add the abbreviation here, CK.

Response to “point 21, statistical analysis”: actually I was refering to the lack of a sub section in M&M describing the statistical analysis that you made, but you added that somewhere else and for me it’s fine.

Figure 6: ok, so now I understand. Fro what you describe you can’t use different clours, otherwise it will be interpreted as up or down. Use a continuious fradient of a single colour,and describe that carefully in the legend. Add the information from “section 4.7. Expression patterns analysis” to the legend of figure 6.
